# Mitigating the Adverse Effects of Polychlorinated Biphenyl Derivatives on Estrogenic Activity via Molecular Modification Techniques

**DOI:** 10.3390/ijerph18094999

**Published:** 2021-05-08

**Authors:** Wei He, Wenhui Zhang, Zhenhua Chu, Yu Li

**Affiliations:** 1MOE Key Laboratory of Resources Environmental Systems Optimization, College of Environmental Science and Engineering, North China Electric Power University, Beijing 102206, China; v493287@163.com (W.H.); zhangwh7722@outlook.com (W.Z.); 2National Center for Climate Change Strategy and International Cooperation, Changsha 410000, China; zhenhua0611@163.com

**Keywords:** polychlorinated biphenyls (PCBs), three-dimensional quantitative structure-activity relationship (3D-QSAR), molecular modifications, molecular dynamics, risk of estrogenic activity

## Abstract

The aim of this paper is to explore the mechanism of the change in oestrogenic activity of PCBs molecules before and after modification by designing new PCBs derivatives in combination with molecular docking techniques through the constructed model of oestrogenic activity of PCBs molecules. We found that the weakened hydrophobic interaction between the hydrophobic amino acid residues and hydrophobic substituents at the binding site of PCB derivatives and human oestrogen receptor alpha (hERα) was the main reason for the weakened binding force and reduced anti-oestrogenic activity. It was consistent with the information that the hydrophobic field displayed by the 3D contour maps in the constructed oestrogen activity CoMSIA model was one of the main influencing force fields. The hydrophobic interaction between PCB derivatives and oestrogen-active receptors was negatively correlated with the average distance between hydrophobic substituents and hydrophobic amino acid residues at the hERα-binding site, and positively correlated with the number of hydrophobic amino acid residues. In other words, the smaller the average distance between the hydrophobic amino acid residues at the binding sites between the two and the more the number of them, and the stronger the oestrogen activity expression degree of PCBS derivative molecules. Therefore, hydrophobic interactions between PCB derivatives and the oestrogen receptor can be reduced by altering the microenvironmental conditions in humans. This reduces the ability of PCB derivatives to bind to the oestrogen receptor and can effectively modulate the risk of residual PCB derivatives to produce oestrogenic activity in humans.

## 1. Introduction

Polychlorinated biphenyls (PCBs) are a type of persistent organic pollutants (POPs) [1,2,3] and, unlike common organic pollutants, they have garnered global attention because of their anti-endocrine activities that have a considerable harmful effect on human health and the environment [4]. Although PCBs do not cause strong and acute toxicity, they are potentially carcinogenic, and cause subacute and chronic poisoning leading to skin damage, cognitive and behavioral impairment, and reproductive and developmental inhibition in both humans [5]. Belonging to the most complex group of endocrine disruptors, PCBs also exhibit strong anti-endocrine activity that can interfere with the oestrogen/androgen system; damage the adipose tissues; and the functions of the reproductive, nervous, and endocrine systems of humans and other organisms are inhibited [6,7,8].

Oestrogens are classified into two types: natural and synthetic. Natural oestrogens include steroids, such as estrone (E_1_) and estradiol (17β-estradiol, E_2_), whereas synthetic oestrogens are non-steroidal hormones. They play a vital role in the growth, development, and reproduction of the human body. Oestrogen can enter cells via transmembrane transport by binding to the corresponding receptors and activate the expression of several genes [9]. Biological activation and detoxification are two ways for the biological transformation of exogenous chemical substances in organisms [10]. After these endocrine disruptors bind to oestrogen receptors, they can not only mimic the effects of natural oestrogen, thereby activating gene expression, but may also prevent natural oestrogen from binding to the corresponding receptors, eventually exhibiting an antagonistic effect on these receptors [11]. PCBs can interact with human oestrogen receptors. Vakharia et al. [12] performed a competitive binding test in human oestrogen receptors to determine the anti-oestrogenic activity of 7 PCBs and their metabolites [13]. In addition, PCBs can also exhibit anti-androgenic activity (similar to the anti-oestrogenic effect) by antagonistically binding to ARs [14]. Bonefeld-Jorgensen et al. [15] studied the anti-androgenic activity of 3 PCBs in Chinese hamster ovary cells through a luciferase reporter gene experiment and found that the anti-androgenic activity of PCB-153 and PCB-180 was significantly less than that of PCB-138. Although PCBs were prohibited from being produced and used worldwide in the 1970s and listed in the Stockholm Convention on Persistent Organic Pollutants [16,17,18], the risk of anti-oestrogenic activity, by a large number of PCBs remaining in the environment, still cannot be ignored. How to effectively regulate and avoid them so as to reduce the damage to human health and environmental pollution is a trending issue that needs to be addressed.

The QSAR (Quantitative structure–activity relationship) method is widely used to study the relationship between the structure and activity data of molecules themselves [19,20]. Chu et al. [21] used the constructed 3D-QSAR (CoMSIA: comparative molecular similarity index analysis) model to predict the anti-oestrogenic activity data of 209 PBDEs’ homologues. In addition, to thoroughly investigate the regulation of the anti-oestrogenic activity of PBDEs, an environmentally friendly PBDE derivative with low anti-oestrogenic activity was designed by combining pharmacophore modelling, fractional analysis experimental design methods, and molecular docking techniques. Li et al. [22] used 12ns molecular dynamics simulation to study the combination of two oestrogen receptors and inhibitors in an aqueous environment, observed changes in the conformation of the system, and found that the anti-oestrogenic activity of the molecules could be effectively modulated by altering human micro-environmental conditions such as the ingestion of compounds capable of inhibiting oestrogenic activity. Considering that PBDEs and PCBs are both POPs and similar to existing studies on the anti-oestrogenic activity of PBDEs, we further modified the design of PCB derivatives, and used molecular docking technology for designing new PCBs derivatives based on the constructed PCB molecular oestrogenic activity model to determine the main factors affecting the interaction of PCBs molecules with oestrogen. Our study provides the theoretical basis and new ideas for regulating and avoiding the risk of anti-oestrogenic activity produced by PCB molecules in the humans.

## 2. Materials and Methods

### 2.1. Construction of a 3D-QSAR Model and Molecular Modification of PCBs with Low Estrogenic Activity

Zhang et al. [23] used the experimental data value pREC_20_ of 11 PCBs measured by the in vitro dual luciferase reporter gene experiment as the 3D-QSAR modelling data. To facilitate subsequent QSAR analysis, the anti-oestrogen activity index of PCBs was calculated as the negative logarithm of the pREC_20_ value.

Among the data values of 11 kinds of PCBs with known pREC_20_ values, PCB-44 with the largest pREC_20_ was selected as the template molecule. Among the data values of 11 PCBs with known pREC_20_ values, PCB-44 with the largest pREC_20_ was selected as the template molecule, seven kinds of PCBs were randomly selected as the training set and four kinds of PCBs were selected as the test set to build a 3D-QSAR model with a 2:1 ratio (ratio of the number of molecules in the training and test sets), and used the Minimize molecular program under SYBYL-X 2.0 to optimize the molecular mechanics of each molecule. The most stable conformation of the molecules was chosen (Table 1) as the lowest energy molecular conformation and the charge carry by the molecule was the Gasteiger-Huckel charge, using the Powell energy gradient method, with a number of iterations of 10,000 and an energy convergence limited to 0.005 kJ·mol^−1^ [24]. The PCB-44 with the largest pREC_20_ value was selected as the modified target molecule, and the force field information of the substitution site and substitution group that had a greater impact on the oestrogenic activity of PCBs molecules was obtained through the 3D contour maps of the CoMSIA model.

### 2.2. Mechanism Analysis Method for Determining the Anti-Oestrogenic Activity of PCBs before and after Modification Based on Molecular Docking

The Surflex-Dock module in SYBYL-X2.0 (Tripos, St. Louis, MO, USA) was used to determine the difference in the effect of docking conformation of PCB molecules with human oestrogen receptor α (hERα) before and after modification. The target molecule PCB-44 and its 7 types of derivative molecules with low anti-oestrogenic activity were docked to each of the five hERα (PDB ID: 2AYR, 2Q70, 1ERR, 2Q6J, and 2IOK [25,26,27,28,29]). Among them, the protein structures of these five hERα were derived from Protein Data Bank (http://www.rcsb.org/pdb, accessed on 25 February 2021). The Tripos force field combined with the minimize module was selected for energy optimization of each molecule to obtain the dominant stable conformation in the lowest energy state combined with the energy gradient method and load MMFF94 charge [30]. The energy convergence limit was set to 0.001 kJ/mol, the maximum optimization time was 10,000, and the others were default values. To expose the structural pockets, it was necessary to remove the ligands, metal ions, and water molecules in the protein receptor molecule before molecular docking, and add polar hydrogen and point charges to it. During the docking process, the threshold and bloat were set to 0.5 and 0, respectively, and other parameters were the default values [31]. After molecular docking, polar interaction, hydrophobic interaction, entropy, and solvation were evaluated together to obtain the corresponding scoring function, which further determined the magnitude of the binding force between the ligand molecule and receptor, with higher scoring values indicating a stronger binding force [32].

## 3. Results and Discussion

The process of this study is shown in Figure 1.

### 3.1. Molecular Substitution Design of Low Anti-Oestrogenic Activity of PCB Derivatives Based on the CoMSIA Model 3D Contour Map

#### 3.1.1. Construction and Verification of the CoMSIA Model of Anti-Oestrogenic Activity of PCBs

The evaluation parameters of the CoMSIA model and the efficiency of the effect of the molecular force field are summarized in Table 2. The model validation method used in this paper is a combination of internal and external validation, where the internal validation uses the LOO cross-validation method, while the external validation is to test the training set model through the test set. The best principal component *n* of the CoMSIA model was 3, and the cross-validation coefficient q^2^ was 0.665 (>0.5), indicating that the model had a reliable predictive capability [33]. In addition, its non-cross-validation r^2^ was 0.991 (>0.8), indicating that the model’s predictive ability was good. The standard deviation SEE was 0.146 and the F value was 114.424, indicating that the two models have a good fit and predictability [34]. The external test coefficient r^2^_pred_ was 0.902, SEP was 0.206, and Q^2^_ext_ was 0.904, indicating that the model has a strong external predictive ability and robustness [35]. In the CoMSIA model, the largest contribution rate of the force field was the electrostatic field (E) of 85.70%, followed by the hydrophobic field (H) of 13.70%, and then the stereo field (S) of 0.50%. The hydrogen bond donor field(D) and hydrogen bond acceptor field (A) did not affect the value of pREC_20_, indicating that the pREC_20_ values of the PCB homologues were affected by the spatial effect, electrical distribution, and hydrophobic interaction between the groups. Electrical distribution and hydrophobic field were the main influencing factors, whereas the hydrogen bond donor–acceptor field had no effect on them.

#### 3.1.2. Analysis of Factors Affecting the Anti-Oestrogenic Activity of PCB Molecules Based on the 3D Contour Map of the CoMSIA Model and the Modification Design of PCB Derivatives with a Low Stimulating Hormone

The 3D contour map of the CoMSIA model is shown in Figure 2. Because the 3D-QSAR model construction needs to select the molecule with the largest effect value [36], PCB-44 with the largest pREC_20_ value was selected as the target molecule. Figure 2a shows the steric field of the CoMSIA model in which the yellow area is mainly located at position 5 of biphenyl, and the green area is mainly located at positions 3 and 5′, indicating that if a smaller volume of the substituent group was substituted into position 5 and a larger volume of the substituent group was substituted into positions 3 and 5′, both help increase the pREC_20_ value of the PCB; namely, the activity of the PCB molecule was enhanced. Figure 2b shows the electrostatic field, where the blue areas are mainly near the carbon skeleton at positions 3′ and 4′ and on position 5, and the red areas are mainly located on positions 3 and 5′, indicating that if the negatively charged substituent group was substituted into positions 3′, 4′, and 5 it helps reduce the pREC_20_ value of the PCBs. Figure 2c represents the hydrophobic field in which the yellow color indicates the favored region. The yellow part was mainly located at position 3, 5′, indicating that the addition of hydrophobic substituents at this position helps increase the activity of PCBs. The disfavored region is indicated in white, and the white region was located at position 5, indicating that the addition of hydrophilic substituents at this position increases the pREC_20_ value of PCBs. The t steric field, electrostatic field, and hydrophobic field, shown in the 3D contour map surround the PCB substitution site, which provided a reference for the selection of low oestrogen activity substitution sites. A comprehensive analysis of the contribution efficiency of each force field in the three-dimensional equipotential diagram of the CoMSIA model to the molecular structure revealed that the three-dimensional field, electrostatic field, and hydrophobic field mainly affect the three substitution sites of Cl_3_, Cl_5_, and Cl_5′_. In other words, introducing small, positively charged or hydrophilic groups at the positions of Cl_3_ and Cl_5′_, and introducing large, negatively charged or hydrophobic groups at the position of Cl_5_ all contributed to the reduction of the pREC_20_ value of the target molecule. Therefore, Cl_3_ and Cl_5′_ were selected as the sites for introducing the substitution groups and 8 types of the most common electronegative groups other than Cl (-Br, -OCH_3_, -C≡CR, -CH=CH_2_, -CH_3_, -CH_2_CH_3_, -CH(CH_3_)_2_, -C(CH_3_)_3_) and 4 common hydrophilic groups (-OH, -CHO, -COOH, -NH_2_) were used as modification groups to produce 50 new PCB-44 molecular derivatives by single and double substitution reactions.

#### 3.1.3. Prediction of the Anti-Oestrogenic Activity of PCB Derivatives and Evaluation of POPs

The pREC_20_ values of 50 types of novel PCB-44 molecular derivatives were predicted using the 3D-QSAR model and the predicted results are summarized in Table 3. The decrease in the pREC_20_ values of PCB-44 derivative molecules ranged from 0.6% to 38.7%; that is, the anti-oestrogenic activity of PCB-44 derivatives was significantly reduced. Among them, the more prominent ones were 5′-tert-butyl-PCB-44 and 3-methyl-5′-methyl-PCB-44, and, compared to PCB-44, their anti-oestrogenic activities were reduced by 38.7% and 37.1%, respectively. This showed that the PCB-44 derivatives designed based on the CoMSIA model had a significant effect on the reduction in anti-oestrogenic activities. We further proved that the change in the structures of PCB molecules can effectively reduce their anti-oestrogenic activity. In addition, the properties of POPs (bio-concentration [37], environmental persistence [38], long-distance migration [39], and biotoxicity [40]) of PCB-44 derivative molecules was evaluated. Our results showed that not only the anti-oestrogenic activity of some derivative molecules decreased significantly but their POP characteristics also decreased to varying degrees. Based on the improvement of estrogenic activity by more than 10%, bio-concentration is reduced by more than 5%, environmental persistence is reduced by more than 15%, the long-distance migration is reduced by more than 5%, and the biotoxicity of new molecules is reduced by more than 0%, 50 types of novel PCB-44 molecular derivatives were screened [41]. After screening, seven novel PCBs molecules with low estrogenic activity were obtained for further study.

### 3.2. Analysis of the Underlying Mechanism That Caused Changes in the Anti-Oestrogenic Activity of PCB Derivatives Based on Molecular Docking Techniques

#### 3.2.1. Analysis of the Molecular Docking Effect of PCB-44 Bound to hERα before and after Modification

In this study, we used the Surflex-Dock module in SYBYL-X2.0 to bind the 7 types of novel and target PCB molecules exhibiting low anti-oestrogenic activity to each of the 5 types of hERα (PDB ID: 2AYR, 2Q70, 1ERR, 2Q6J, and 2IOK [25,26,27,28,29]), and the docking scores are given in Table 4. A previous study reported that the effect of oestrogen on gene expression was mainly determined by the binding of the target molecule and hERα [42]. Therefore, by analyzing the binding effect of the target molecule and hERα, the anti-oestrogenic activity of the target molecule can be more accurately evaluated [43]. Therefore, the oestrogen activity of PCB derivatives bound to different hERα was preliminarily evaluated, according to the docking scoring function (binding force) of the new PCB molecules and hERα (Table 4).

As summarized in Table 4, when the 7 types of PCB-44 derivatives were docked to the 2AYR oestrogen active receptor protein, the docking score of the 3-amino-PCB-44 molecule decreased, whereas that of others increased. Therefore, we concluded that 3-amino-PCB-44 can effectively inhibit the expression of the oestrogenic activity of the 2AYR gene in endocrine glands and reduce its anti-oestrogenic activity. Analyzing the results of the docking to the 1ERR protein, the docking scores of 3-amino-PCB-44 and 3-amino-5′-amino-PCB-44 decreased, indicating that these two novel PCB-44 molecules could effectively inhibit the activity of the 1ERR protein, thus making it less likely to exhibit anti-oestrogenic activity in the endocrine gland. The docking results of the 3 types of hERα enzymes 2Q70, 2Q6J, and 2IOK showed that the docking scores of 3-methyl-PCB-44 were lower than those of their target molecules, while the docking scores of 6 novel molecules, 3-amino-PCB-44, 3-vinyl-5′-amino-PCB-44, 3-methyl-5′-amino-PCB-44, 3-isopropyl-5′-amino-PCB-44, 3-amino-5′-amino-PCB-44, and 3-hydroxy-5′-hydroxy-PCB-44, increased to varying degrees compared to the template molecules. We therefore speculated that the designed 3-methyl-PCB-44 binds to the three proteins 2Q70, 2Q6J, and 2IOK to a lesser extent, thereby inhibiting the activities of the 3 hERα receptor proteins.

Our analysis showed that 3-amino-PCB-44, 3-amino-5′-amino-PCB-44, and 3-methyl-PCB-44 ensured a significant reduction in the properties of POPs along with a significant reduction in binding to the oestrogen receptor. When 3-amino-PCB-44, 3-amino-5′-amino-PCB-44 and 3-methyl-PCB-44 molecules docked to the hERα receptor protein, amino acid residues played a major role, they surrounded the protein and were active residues, at a certain distance from the protein. Therefore, we considered 3-methyl-PCB-44 docking to two hERα receptor proteins as the target molecule (Figure 2) to analyze the mechanism of the interaction between active residues and the 5 oestrogen receptor proteins. The specific amino acid distribution is shown in Figure 3.

As shown in Figure 3, hydrophobic residues, such as Leu391, Leu387, Met388, Trp383, Leu384, Ile424, Leu525, Leu346, Ala350, Leu349, and Phe404 surrounded the PCB-44 binding site to hERα (PDB ID: 2Q70) and these residues together formed a hydrophobic binding cavity that interacted with the hydrophobic -Cl substituent resulting in stronger binding, whereas the hydrophilic amino acid residue Arg394 was farther away from each hydrophobic group and had less effect on the strong hydrophobic interaction between the hydrophobic group and PCB-44. 3-Methyl-PCB-44 was surrounded by hydrophobic residues, such as Leu428, Met388, Leu391, Leu387, Leu384, Phe404, Leu346, Ala350, Met343, and Leu525 around the binding site of hERα (PDB ID: 2Q70). The Thr347 and His524 hydrophilic groups inhibited the hydrophobic interaction between the -Cl substituent and the surrounding hydrophobic groups. As the result, the binding effect between Thr347 and His524 was weakened, which can explain the lower anti-oestrogenic activity of 3-methyl-PCB-44 mediated by hERα (PDB ID: 1ERR) (Figure 4) than that of PCB-44.

Chu et al. [44] studied the molecular docking results of PBDEs and their metabolites via hERα binding and deduced that the main factor affecting the expression of anti-oestrogenic activity of PBDEs and their metabolites was the hydrophobic interaction between PBDE molecules and oestrogenic receptors that were used to perform the study of anti-oestrogenic activity of PCB molecules and also further confirmed our result that the hydrophobic interaction between PCB molecules and hERα could significantly affect the expression of anti-oestrogenic activity of PCB molecules. In addition, the hydrophobic/hydrophilic balance is the main feature of the protein structure, and the hydrophobic interaction within the molecule is the main factor affecting the stability of its structure [45]. The hydrophobic amino acid residues and hydrophobic substituents at the binding site of PCB molecules and hERα had a hydrophobic effect, which lead to the enhancement of the binding force of t PCBs molecules and hERα, thus resulting in the significant expression of anti-oestrogenic activity of PCB molecules.

#### 3.2.2. Analysis of Amino Acid Residues Docked to hERα before and after Molecular Modification of PCBs

The relationship between the hydrophobic interaction and the binding force of the novel molecules of PCBs and hERα was further investigated by using molecular docking techniques, and using PCB-44 molecules before and after modification docking to hERα (PDB ID: 2Q6J), respectively. The results are given in Figure 5. The key hydrophobic amino acid residues at the active site of hERα (PDB ID: 2Q6J) were obtained by using the National Centre for Biotechnology Information (NCBI) query for Leu 346, Ala 350, Leu387, Met388, Phe404, Met421, Met388, Phe404, and Met421, and the average distance between the hydrophobic substituents of the PCB molecules before and after modification and the hydrophobic amino acid residues of hERα (PDB ID: 2Q6J) were measured separately (Table 5).

As summarized in Table 5, the average distance between the substituent of PCB-44 and the hydrophobic amino acid residues Leu346, Ala350, Leu387, Met388, Phe404, and Met421 at the active site of hERα (PDB ID: 2Q6J) was 5.23 Å. Using this value as a benchmark, comparative analysis of the average distance between the novel PCB-44 molecule and the hydrophobic amino acid residues at the active site of hERα (PDB ID: 2Q6J) revealed that the average distance between the vinyl substituent at neighboring position 3 of the 3-vinyl-5′-amino-PCB-44 molecule and the hydrophobic amino acid residues was 5.02 Å (<5.23 Å). The 3-methyl-5′-amino-PCB-44 molecule had an average distance of 5.07 Å (<5.23 Å) between the methyl substituent in the neighboring 3 position and the hydrophobic ammonia residue. The combined docking scores of 3-vinyl-5′-amino-PCB-44 (3.22) and 3-methyl-5′-amino-PCB-44 (3.63) docked to hERα (PDB ID: 2Q6J) (Table 5) were all higher than the docking scores of PCB-44 (2.94). Therefore, we inferred that the average distance between the hydrophobic substituent and the hydrophobic amino acid residue was significantly and inversely related to the strength of the hydrophobic interaction between the two, signifying greater binding force between them. Wertz and Scheraga et al. [46] showed that the magnitude of the hydrophobic effect was related to the fraction of each residue buried inside the molecule (Pin), which was affected by the total number of amino acid residues, thus indicating that the magnitude of the hydrophobic effect is affected by the total number of amino acids. Therefore, although the average distance of 3-methyl-PCB-44 molecules is 4.2 (<5.23 Å), its docking score (2.57) was still lesser than that of PCB-44 (2.94), which may be related to its surrounding residues. When the number of hydrophobic residues was the same, the binding force between PCB molecules and hERα decreased as the average distance between hydrophobic substituents and hydrophobic amino acid residues increased, thereby inhibiting the expression of anti-oestrogenic activity of PCB molecules in the human environment.

In summary, the fewer the hydrophobic residues surrounding the new PCB molecules, the less hydrophobic the interaction between them, and the more loosely bound they are, the lower anti-oestrogenic activity is expressed by the PCB molecules. The lesser the average distance between the hydrophobic substituents of PCB derivatives and the hydrophobic amino acid residues of the oestrogen receptor-binding site, the stronger the hydrophobic interaction between the two. That is, the stronger the binding force between the PCB molecule and the oestrogen receptor, the stronger the anti-oestrogenic activity of the PCB molecule, thus further explaining the differential anti-oestrogenic activity of the PCB-44 derivative molecule when bound to the oestrogen receptor.

### 3.3. Validation in Oestrogenic Effect Mitigation of PCBs Based on Molecular Dynamics Simulations

We further screened the conditions of ingested substances in vivo conditions of the human body that could be adjusted to reduce the effect of PCBs before and after modification to bind to oestrogen receptors in the human body in order to prevent the expression of oestrogen activity. Thus, we created a partial factorial design of 7 PCB derivatives with reduced anti-oestrogenic activity using molecular dynamics assisted by the L9(3^4^) Taguchi experiment [47] to calculate the binding energy values of PCB-44 and its derivative molecules to dock the oestrogen receptor protein 2Q6J enzyme under the conditions of 9 sets of experimental protocols. A previous study [48] showed that polymers containing hydrophobic side chains or hydrophobic blocks were used to control the hydrophobic interaction to adjust the interaction with proteins. Compounds found in common foods, such as xylitol (yoghurt, chewing gum), mannitol [49] (pumpkin, mushroom, onion, and kelp), tannins [50] (apple, banana, persimmon, and grape), and anthocyanin [51] (grape, prune, mulberry, kale, and purple potato) bound to proteases and effectively alter the conformation and microenvironment of the protein, thus enhancing the polarity of the human microenvironment, in which tryptophan or tyrosine residues are located, thereby weakening the hydrophobic effect of the bound conformation. Therefore, the experimental conditions set in this study were mainly the concentration of xylitol (A), mannitol (B), tannins (C), and anthocyanin (D), simulated with reference to human environmental conditions, and three concentration levels of each factor were set at low, medium, and high levels [52] (the 4 factors and 3 levels of combination scheme are given in Table 6). Factorial analysis was performed to determine the optimal level combination that effectively reduced the anti-oestrogen activity and intake conditions of PCB derivatives. The results of the molecular dynamics simulation binding energy of PCBs before and after modification are given in Table 7. The binding energies of the oestrogen receptor enzyme to the target molecule PCB-44 and 7 novel derivative molecules were calculated under optimal conditions. We explored the effect of reducing oestrogen activity before and after PCB-44 molecular modification in the real environment by aiming to combine the internal modification of the molecules with external conditions to mitigate the risk of anti-oestrogenic activity of residual PCBs in humans.

Previous studies [53] have shown that the smaller the absolute value of the binding energy, the weaker the affinity of the molecule to bind to the receptor protein; namely, the molecule exhibits lower anti-oestrogenic activity. As explained in Table 7, under the same external conditions of the 7 types of PCB-44 derivatives, only the absolute value of the binding energy of PCB-44-6 and PCB-44-7 to the oestrogen receptor showed a decreasing trend; the reduction rates were 13.77% to 33.66% and 19.99% to 31.68%, respectively, indicating that the anti-oestrogenic activity of these two molecules was expressed to a much lesser extent than the standard molecule PCB-44. The anti-oestrogenic activity of these two molecules was much lower than that of PCB-44. Therefore, 2 new types of PCB-44 derivatives with low anti-oestrogenic activity (PCB-44-6 and PCB-44-7) were screened, and they are more difficult to bind to estrogen receptors in the microenvironment of different components of the ingested substances. By comparing the binding energies of PCB-44-6 and PCB-44-7 derivatives docked to the 2Q6J enzyme under the effect of different combinations of component conditions (Figure 6), the absolute value of the binding energy of PCB-44-6 and PCB-44-7 derivatives docked to the 2Q6J enzyme was minimized under the component conditions of combination 4 (xylitol concentration of 0.95 mmol/L, mannitol concentration of 1.92 mmol/L, tannic acid concentration of 0.94 mmol/L and anthocyanin concentration of 1.92 mmol/L). It showed that the anti-oestrogenic activity of PCB-44-6 and PCB-44-7 expressed under the effect of the combination 4 conditions was the most significant, and it was the optimal intake condition combination to inhibit the expression of anti-oestrogen activity of PCB derivatives.

## 4. Conclusions

We used the 3D-QSAR model combined with molecular docking technology to investigate in detail the mechanism of differential expression of anti-oestrogenic activity of PCB molecules before and after modification, which was significantly and positively correlated with the hydrophobic interaction between PCB molecules and oestrogenic receptors. The main factor affecting the hydrophobic interaction between PCBs before and after modification and the oestrogen receptor was the number of hydrophobic amino acid residues and the average distance between them as well as the hydrophobic substituent; the intake of xylitol, mannitol, tannins, and anthocyanin in certain proportions significantly attenuated the hydrophobic effect of the microenvironment in which the amino acid residues of PCB molecules bound to oestrogenic receptors by using molecular dynamics simulations assisted by Taguchi’s experimental design. That is, adjusting the human microenvironment can indirectly affect the expression of estrogen activity of PCBs molecules exposed to the human environment, and alleviate the risk of estrogen activity of residual PCBs molecules in the human body. In this paper, the main factors affecting the estrogenic activity of PCBs molecules and the designed adjustment scheme of human intake microenvironment are determined, which provide a theoretical basis and new ideas for the risk of estrogenic activity produced by PCBs molecules.

## Figures and Tables

**Figure 1 ijerph-18-04999-f001:**
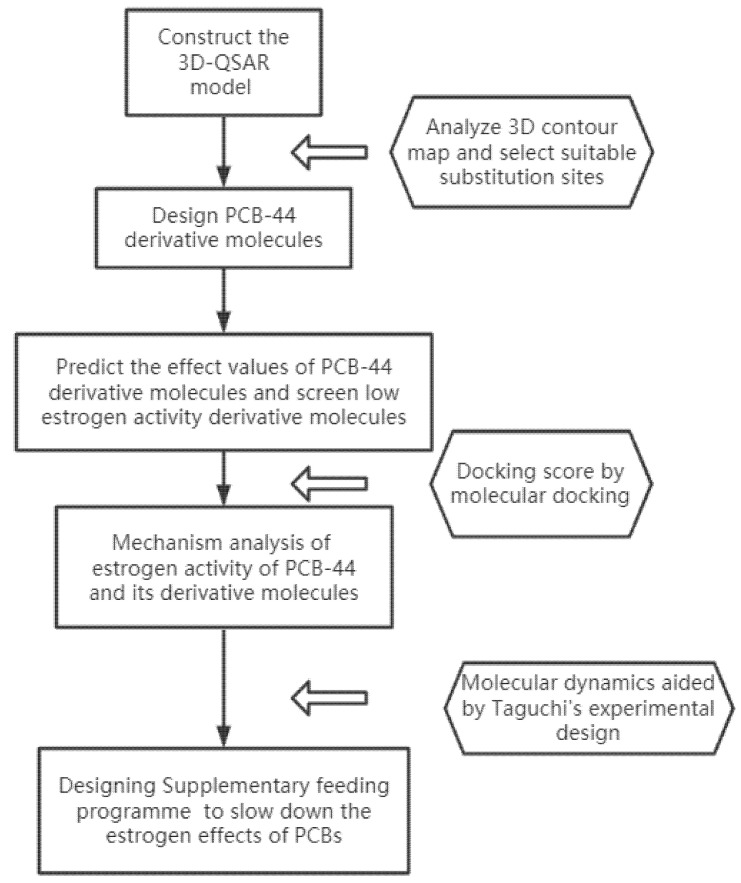
Flow chart of the risk mechanism of PCBs estrogenic activity and its regulation based on molecular modifications.

**Figure 2 ijerph-18-04999-f002:**
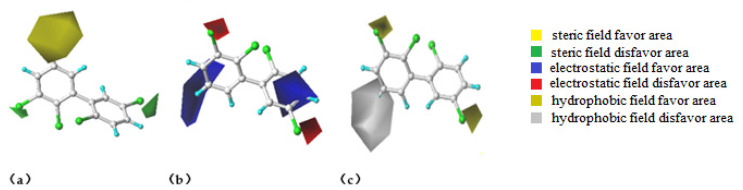
A 3D contour map of the CoMSIA steric field (**a**), electrostatic field (**b**), and hydrophobic field (**c**) of PCB-44.

**Figure 3 ijerph-18-04999-f003:**
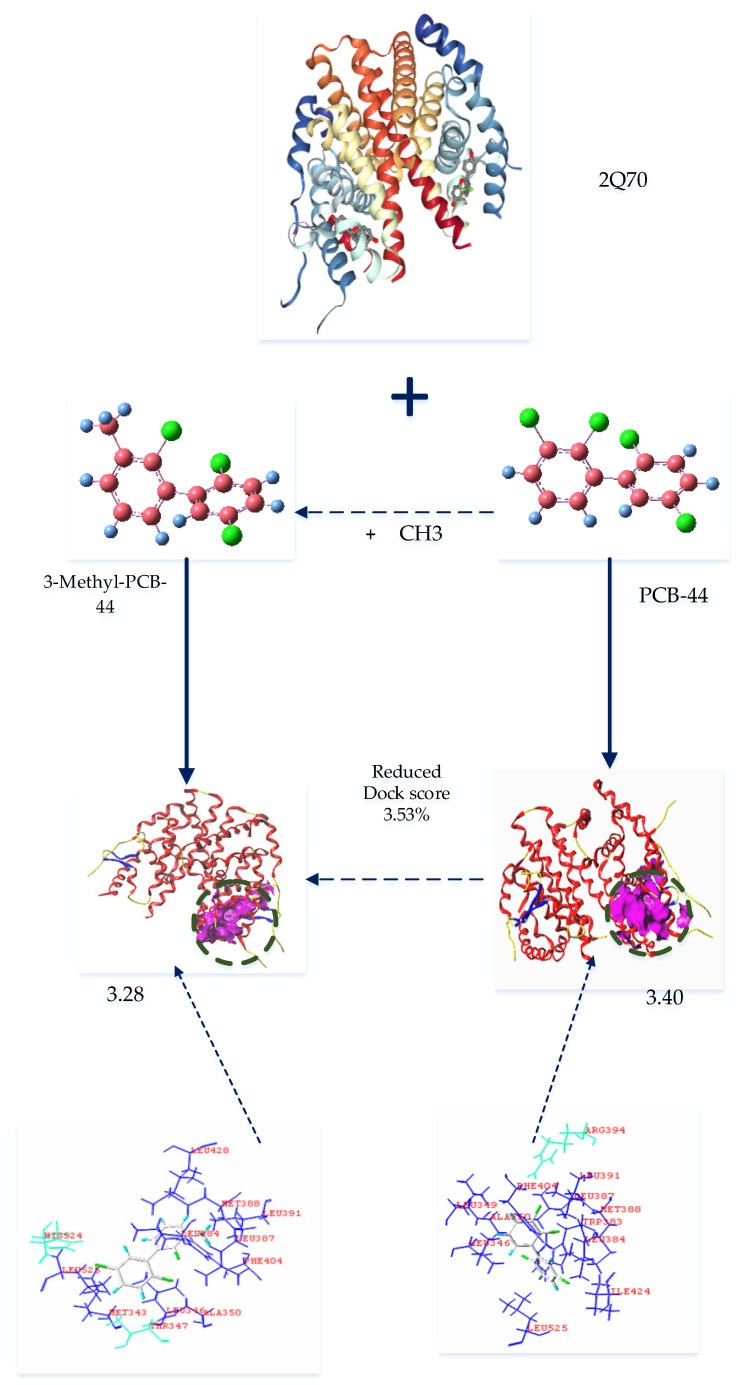
Binding conformation of 3-methyl-PCB-44 to 2Q70 receptor protein.

**Figure 4 ijerph-18-04999-f004:**
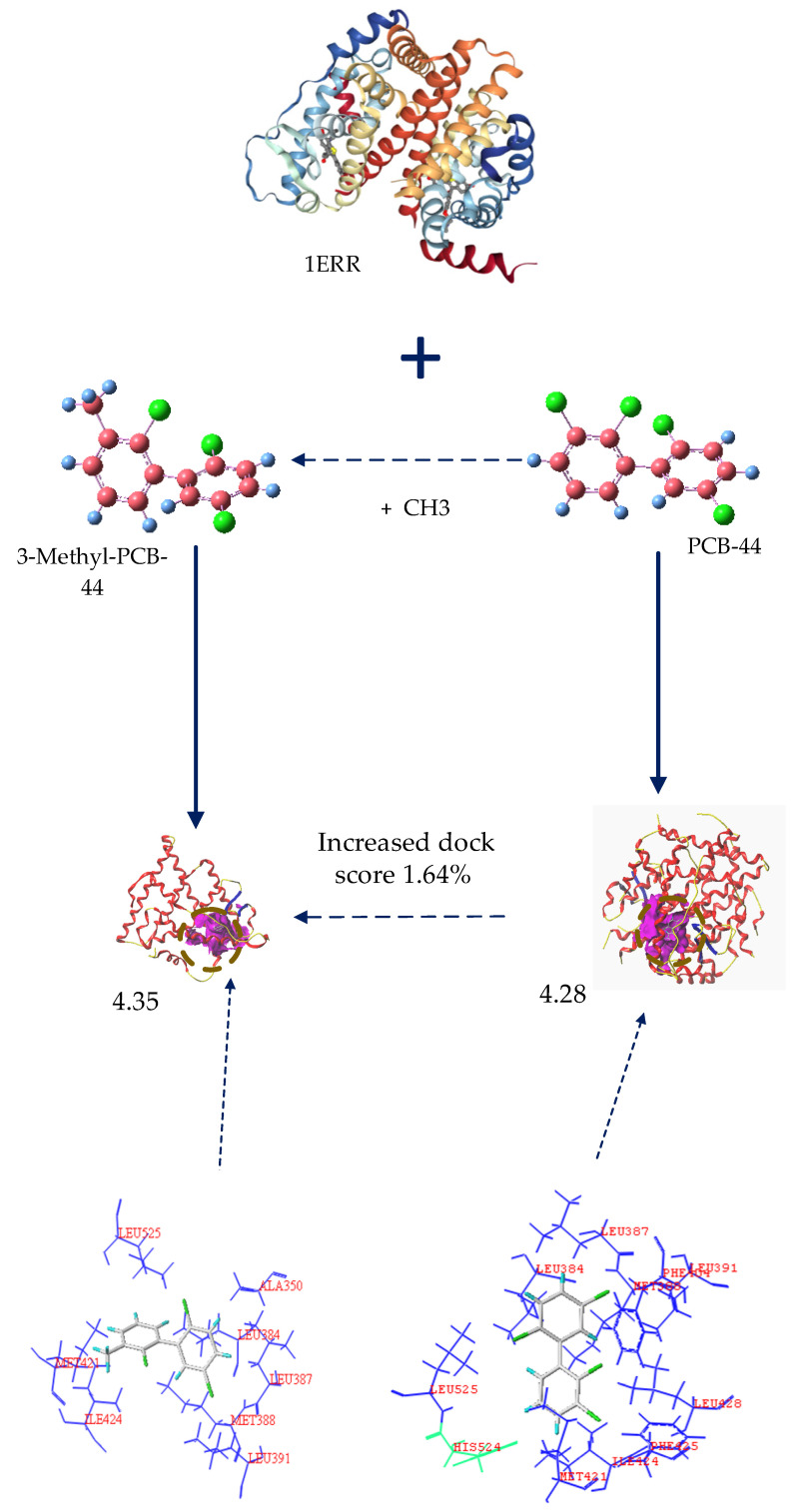
Binding conformation of 3-methyl-PCB-44 to 1ERR receptor protein.

**Figure 5 ijerph-18-04999-f005:**
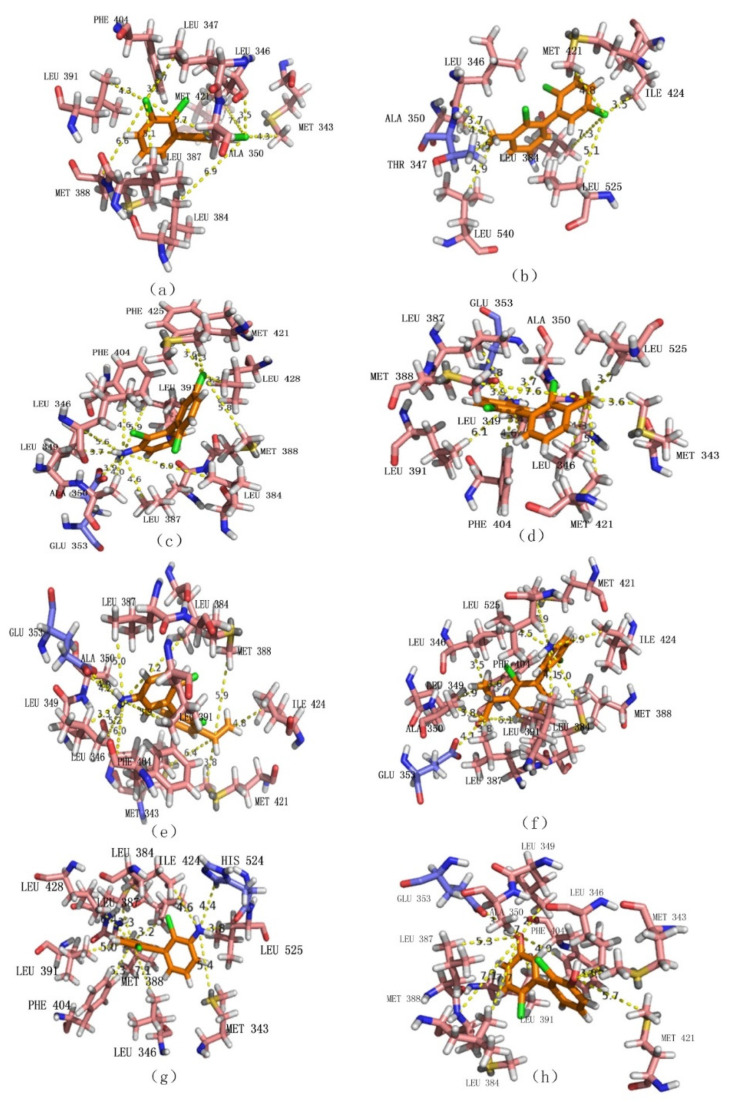
Docking conformation of PCB-44 molecule (**a**) and the modified novel molecule 3-methyl-PCB-44 (**b**), 3-amino-PCB-44 (**c**), 3-methyl-5′-amino-PCB-44 (**d**), 3-vinyl-5′-amino-PCB-44 (**e**), 3-isopropyl-5′-amino-PCB-44 (**f**), 3-amino-5′-amino-PCB-44 (**g**)**,** and 3-hydroxy-5′-hydroxy-PCB-44 (**h**) docked to hERα (PDB ID: 2Q6J), respectively. (Orange rod-shaped structure represents the PCB-44 molecule; blue and pink rod-shaped structures represent hydrophilic and hydrophobic amino acid residues at the binding site, respectively; yellow dashed line represents the distance between the hydrophobic substituent and the hydrophobic amino acid residue of PCB-44 molecules).

**Figure 6 ijerph-18-04999-f006:**
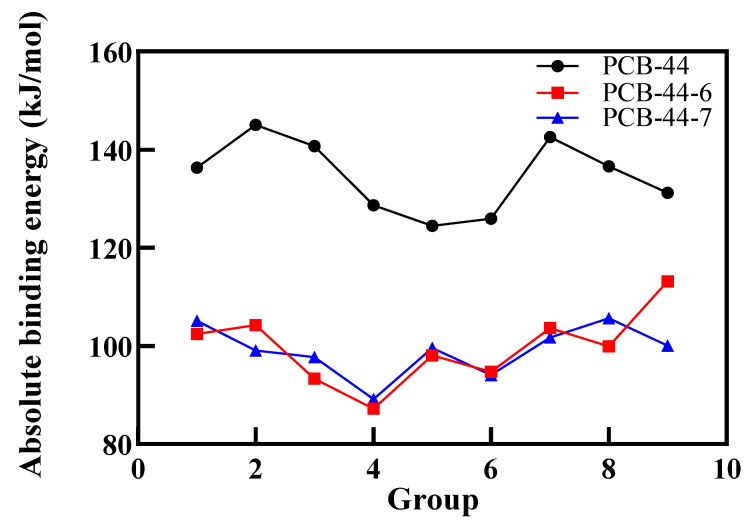
Binding energy of PCB-44, PCB-44-6, and PCB-44-7 molecules docked to oestrogen receptor binding under different combinations of components.

**Table 1 ijerph-18-04999-t001:** Observed pREC_20_ values of training set and test set molecules on the CoMSIA model.

No.	Compounds	Observed Value
1^a^	2,2′,5-Trichlorobiphenyl (PCB-18)	5.006
2 ^a^	2,4,6-Trichlorobiphenyl (PCB-30)	6.614
3 ^a^	2,2′,3,5′-Tetrachlorobiphenyl (PCB-44)	8.158
4 ^a^	2,2′,5,5′-Tetrachlorobiphenyl (PCB-52)	6.509
5 ^a^	2,2′,4,4′,5-Pentachlorobiphenyl (PCB-99)	6.048
6 ^a^	2,2′,4,5,5′-Pentachlorobiphenyl (PCB-101)	7.189
7 ^a^	2,3,3′,4′,6-Pentachlorobiphenyl (PCB-110)	8.009
8 ^b^	2,2′,3,3′,4,4′-Hexachlorobiphenyl (PCB-128)	7.073
9 ^b^	2,2′,4,5′,6-Pentachlorobiphenyl (PCB-103)	7.770
10 ^b^	2,2′,4,5′-Tetrachlorobiphenyl (PCB-49)	6.672
11 ^b^	2,4,4′-Trichlorobiphenyl (PCB-28)	5.430

^a^: Training set; ^b^: test set.

**Table 2 ijerph-18-04999-t002:** Evaluation parameters of the CoMSIA model and the contribution rate of the molecular force field to the pREC_20_ values of PCBs.

Model	*n*	q^2^	r^2^	SEE	F	r^2^_pred_	SEP	Q^2^_ext_	S	E	H	D	A
CoMSIA	3	0.665	0.991	0.146	114.424	0.902	0.206	0.904	0.50%	85.70%	13.70%	0	0

*n* is the optimal number of principal components, q^2^ is the cross-validation coefficient, r^2^ is the non-cross-validation coefficient, SEE is the standard deviation, F is the F-test value, r^2^_pred_ is the external test coefficient, SEP is the standard error of prediction, and Q^2^_ext_ is the external sample calibration complex correlation coefficient, E is the electrostatic field, H is the hydrophobic field, S is the stereo field, D is the hydrogen bond donor field, A is the hydrogen bond acceptor field.

**Table 3 ijerph-18-04999-t003:** Predicted anti-oestrogenic activity, bio-concentration, environmental persistence, long-distance migration, and biological toxicity of the novel PCB-44 molecules.

Compounds	pREC_20_		logBCF [33]		log*t*_1/2_ [36]		log*K*_OA_ [37]		pEC_50_ [38]	
	CoMSIA Pred.	Fall Rate (%)	CoMSIA	Change Rate (%)	CoMSIA	Change Rate (%)	CoMSIA	Change Rate (%)	CoMFA	Change Rate (%)
Target molecule PCB-44	8.158		4.542		0.94		9.318		4.327	
3-carboxyl-PCB-44	7.712	5.5	4.204	−7.44	0.830	−11.70	8.731	−6.30	4.878	12.73
3-amino-PCB-44	7.009	14.1	4.042	−11.01	0.694	−26.17	8.210	−11.89	3.943	−8.87
3-vinyl-PCB-44	5.201	36.2	4.258	−6.25	0.695	−26.06	8.459	−9.22	4.774	10.33
3-methyl-PCB-44	6.991	14.3	4.036	−11.14	0.704	−25.11	8.201	−11.99	4.153	−4.02
3-ethyl-PCB-44	7.200	11.7	4.040	−11.05	0.709	−24.57	8.291	−11.02	4.423	2.22
3-isopropyl-PCB-44	7.353	9.9	4.143	−8.78	0.710	−24.47	8.370	−10.17	4.401	1.71
5′-carboxyl-PCB-44	5.260	35.5	4.817	6.05	1.075	14.36	9.770	4.85	3.243	−25.05
5′-amino-PCB-44	6.459	20.8	3.757	−17.28	0.546	−41.91	7.990	−14.25	4.641	7.26
5′-hydroxyl-PCB-44	7.637	6.4	4.367	−3.85	0.832	−11.49	8.863	−4.88	4.754	9.87
5′-vinyl-PCB-44	5.881	27.9	4.354	−4.14	0.740	−21.28	8.667	−6.99	4.688	8.34
5′-methyl-PCB-44	6.957	14.7	4.936	8.67	1.206	28.30	9.181	−1.47	5.171	19.51
5′-ethyl-PCB-44	6.616	18.9	4.230	−6.87	0.601	−36.06	8.030	−13.82	4.728	9.27
5′-isopropyl-PCB-44	6.094	25.3	4.169	−8.21	0.667	−29.04	8.321	−10.70	4.657	7.63
5′-tert-butyl-PCB-44	5.000	38.7	4.998	10.04	1.046	11.28	9.826	5.45	5.235	20.98
3-Br-5′-aldehyde-PCB-44	6.838	16.2	4.845	6.67	1.177	25.21	9.096	−2.38	5.171	19.51
3-Br-5′-amino-PCB-44	7.057	13.5	4.314	−5.02	0.819	−12.87	8.790	−5.67	4.869	12.53
3-Br-5′-hydroxyl-PCB-44	7.581	7.1	4.310	−5.11	0.796	−15.32	8.758	−6.01	4.752	9.82
3-methoxy-5′-aldehyde-PCB-44	5.698	30.2	5.563	22.48	0.901	−4.15	9.292	−0.28	3.186	−26.37
3-methoxy-5′-carboxyl-PCB-44	5.690	30.3	5.483	20.72	0.969	3.09	9.623	3.27	4.043	−6.56
3-methoxy-5′-amino-PCB-44	6.665	18.3	2.559	−43.66	0.458	−51.28	7.816	−16.12	5.187	19.88
3-methoxy-5′-hydroxyl-PCB-44	5.682	30.4	4.727	4.07	0.743	−20.96	8.813	−5.42	3.999	−7.58
3-ethinyl-5′-aldehyde-PCB-44	5.676	30.4	5.220	14.93	0.939	−0.11	9.188	−1.40	2.945	−31.94
3-ethinyl-5′-carboxyl-PCB-44	5.669	30.5	5.133	13.01	1.007	7.13	9.518	2.15	3.795	−12.29
3-ethinyl-5′-amino-PCB-44	6.503	20.3	3.652	−19.59	0.494	−47.45	7.863	−15.61	4.595	6.19
3-ethinyl-5′-hydroxyl-PCB-44	6.084	25.4	2.715	−40.22	0.840	−10.64	7.970	−14.47	4.424	2.24
3-vinyl-5′-aldehyde-PCB-44	7.783	4.6	4.818	6.08	0.960	2.13	8.803	−5.53	4.606	6.45
3-vinyl-5′-carboxyl-PCB-44	5.290	35.2	4.929	8.52	0.877	−6.70	8.983	−3.60	3.901	−9.85
3-vinyl-5′-amino-PCB-44	5.629	31	2.585	−43.09	0.548	−41.70	6.910	−25.84	4.070	−5.94
3-vinyl-5′-hydroxyl-PCB-44	7.167	12.1	3.911	−13.89	0.667	−29.04	8.265	−11.30	4.596	6.22
3-methyl-5′-aldehyde-PCB-44	6.950	14.7	4.755	4.69	0.889	−5.43	8.277	−11.17	4.576	5.75
3-methyl-5′-carboxyl-PCB-44	7.714	5.4	4.801	5.70	1.006	7.02	8.819	−5.36	3.853	−10.95
3-methyl-5′-amino-PCB-44	6.202	24	2.687	−40.84	0.441	−53.09	6.455	−30.73	4.017	−7.16
3-methyl-5′-hydroxyl-PCB-44	6.343	22.2	3.859	−15.04	0.596	−36.60	7.747	−16.86	4.584	5.94
3-ethyl-5′-aldehyde-PCB-44	7.135	12.5	4.993	9.93	0.893	−5.00	8.378	−10.09	4.750	9.78
3-ethyl-5′-carboxyl-PCB-44	7.893	3.2	5.041	10.99	1.010	7.45	8.918	−4.29	4.026	−6.96
3-ethyl-5′-amino-PCB-44	5.341	34.5	3.491	−23.14	0.313	−66.70	6.967	−25.23	4.625	6.89
3-ethyl-5′-hydroxyl-PCB-44	6.524	20	4.090	−9.95	0.600	−36.17	7.844	−15.82	4.749	9.75
3-isopropyl-5′-aldehyde-PCB-44	7.272	10.9	4.754	4.67	0.894	−4.89	8.421	−9.63	4.683	8.23
3-isopropyl-5′-carboxyl-PCB-44	8.034	1.5	4.804	5.77	1.011	7.55	8.962	−3.82	3.960	−8.48
3-isopropyl-5′-amino-PCB-44	5.917	27.5	2.615	−42.43	0.454	−51.70	6.559	−29.61	4.098	−5.29
3-isopropyl-5′-hydroxyl-PCB-44	6.665	18.3	3.865	−14.91	0.601	−36.06	7.889	−15.34	4.684	8.25
3-tert-butyl-5′-amino-PCB-44	7.539	7.6	4.409	−2.93	0.617	−34.36	8.620	−7.49	5.426	25.40
3-Br-5′-Br-PCB-44	8.106	0.6	4.406	−2.99	0.863	−8.19	9.081	−2.54	4.347	0.46
3-carboxyl-5′-carboxyl-PCB-44	5.726	29.8	5.193	14.33	1.277	35.85	10.435	11.99	2.538	−41.35
3-amino-5′-amino-PCB-44	5.182	36.5	3.276	−27.87	0.300	−68.09	6.882	−26.14	4.246	−1.87
3-hydroxyl-5′-hydroxyl-PCB-44	5.809	28.8	4.255	-6.32	0.762	−18.94	8.407	−9.78	3.549	−17.98
3-vinyl-5′-vinyl-PCB-44	5.333	34.6	4.032	−11.23	0.570	−39.36	7.996	−14.19	4.543	4.99
3-methyl-5′-methyl-PCB-44	5.131	37.1	3.440	−24.26	0.346	−63.19	6.804	−26.98	4.406	1.83
3-ethyl-5′-ethyl-PCB-44	5.530	32.2	3.544	−21.97	0.370	−60.64	7.003	−24.84	4.801	10.95
3-isopropyl-5′-isopropyl-PCB-44	5.834	28.5	4.542	0.00	0.394	−58.09	7.194	−22.79	4.327	0.00

pREC_20_: logarithm of estrogen activity indicators; logBCF: logarithm of bioconcentration factors; log*t*_1/2_: logarithm of biological half-life; pEC_50_: logarithm of concentration for 50% of maximal effect.

**Table 4 ijerph-18-04999-t004:** Docking scores of 7 types of novel PCBs molecules bound to different types of hERα.

Compounds	2AYR		2Q70		1ERR		2Q6J		2IOK	
	Docking Score	Change Rate (%)	Docking Score	Change Rate (%)	Docking Score	Change Rate (%)	Docking Score	Change Rate (%)	Docking Score	Change Rate (%)
Target molecule PCB-44	3.06		3.40		4.28		2.94		2.90	
3-amino-PCB-44	2.74	−10.46	3.57	5.00	3.67	−14.25	3.41	16.10	3.49	20.39
3-methyl-PCB-44	4.22	37.91	3.28	−3.53	4.35	1.64	2.57	−12.61	2.82	−2.84
3-vinyl-5′-amino-PCB-44	5.05	65.03	5.86	72.35	4.30	0.47	3.22	9.74	4.29	47.88
3-methyl-5′-amino-PCB-44	4.23	38.24	5.20	52.94	5.84	36.45	3.63	23.70	4.76	64.34
3-isopropyl-5′-amino-PCB-44	5.11	66.99	5.78	70.00	4.38	2.34	4.04	37.45	4.81	65.93
3-amino-5′-amino-PCB-44	4.39	43.46	5.54	62.94	3.50	−18.22	4.03	37.13	5.43	87.41
3-hydroxyl-5′-hydroxyl-PCB-44	4.10	33.99	4.38	28.82	4.95	15.65	4.45	51.48	4.59	58.43

**Table 5 ijerph-18-04999-t005:** Distance between the hydrophobic substituents of PCB-44 and its new molecules and the hydrophobic amino acid residues at the binding site of hERα (PDB ID: 2Q6J).

	Before Modification	After Modification						
Name of Amino Acid Residue	Target Molecule PCB-44	3-methyl-PCB-44	3-amino-PCB-44	3-hydroxyl-5′-hydroxyl-PCB-44	3-vinyl-5′-amino-PCB-44	3-methyl-5′-amino-PCB-44	3-isopropyl-5′-amino-PCB-44	3-amino-5′-amino-PCB-44
Docking score	2.94	2.57	3.41	4.45	3.22	3.63	4.04	4.03
Hydrophobic residues	Leu346	3.5	3.7	5.6	3.9	5.2	4.3	3.5
	Ale350	5.7	4.1	4.0	4.7	4.2	3.7	3.8
	Leu387	5.1	-	4.6	5.3	5.0	4.8	3.8
	Met388	6.6	-	5.8	7.1	5.9	7.6	5.0
	Phe404	3.1	-	4.6	4.0	6.0	4.6	3.6
	Met421	7.4	4.8	4.3	5.7	3.8	5.4	6.9
Average distance (Å)	5.23	4.20	4.82	5.12	5.02	5.07	4.43	5.00
Hydrophilic residues	Thr347	-	3.5	-	-	-	-	-
	Glu353	-	-	3.9	3.9	4.0	3.9	4.1
	His524	-	-	-	-	-	-	-

**Table 6 ijerph-18-04999-t006:** Four factors and 3 levels of combination scheme of external conditions of molecular dynamics.

Combinations	A (mmol/L)	B (mmol/L)	C (mmol/L)	D (mmol/L)
1	0	0	0	0
2	0	0.96	0.94	0.96
3	0	1.92	1.88	1.92
4	0.95	0	0.94	1.92
5	0.95	0.96	1.88	0
6	0.95	1.92	0	0.96
7	1.90	0	0.94	0.96
8	1.90	0.96	0	1.92
9	1.90	1.92	1.88	0

**Table 7 ijerph-18-04999-t007:** Effect of the same external conditions on the binding of PCB-44 molecule docked to oestrogen receptor before and after modification.

	PCB-44	3-methyl-PCB-44	3-amino-PCB-44	3-hydroxyl-5′-hydroxyl-PCB-44	3-vinyl-5′-amino-PCB-44	3-methyl-5′-amino-PCB-44	3-isopropyl-5′-amino-PCB-44	3-amino-5′-amino-PCB-44
Combination 1	−136.317	−133.293	−138.483	−128.696	−135.967	−140.651	−102.431	−105.153
Change rate (%)		−2.22	1.59	−5.59	−0.26	3.18	−24.86	−22.86
Combination 2	−145.044	−139.843	−107.763	−115.925	−128.898	−129.797	−104.278	−99.087
Change rate (%)		−3.59	−25.7	−20.08	−11.13	−10.51	−28.11	−31.68
Combination 3	−140.755	−134.72	−131.681	−127.886	−124.711	−151.602	−93.373	−97.705
Change rate (%)		−4.29	−6.45	−9.14	−11.4	7.71	−33.66	−30.59
Combination 4	−128.654	−135.583	−136.997	−112.013	−132.642	−136.842	−87.189	−89.155
Change rate (%)		5.39	6.48	−12.93	3.1	6.36	−32.23	−30.7
Combination 5	−124.468	−126.849	−129.241	−124.585	−121.917	−133.491	−98.075	−99.583
Change rate (%)		1.91	3.83	0.09	−2.05	7.25	−21.2	−19.99
Combination 6	−125.964	−137.308	−119.652	−129.574	−131.267	−140.547	−94.746	−94.105
Change rate (%)		9.01	−5.01	2.87	4.21	11.58	−24.78	−25.29
Combination 7	−142.61	−127.914	−120.375	−116.144	−129.26	−137.648	−103.637	−101.748
Change rate (%)		−10.31	−15.59	−18.56	−9.36	−3.48	−27.33	−28.65
Combination 8	−136.574	−127.681	−119.583	−126.673	−106.3	−126.598	−99.966	−105.671
Change rate (%)		−6.51	−12.44	−7.25	−22.17	−7.3	−26.8	−22.63
Combination 9	−131.23	−107.546	−126.97	−123.287	−132.44	−138.777	−113.162	−100.038
Change rate (%)		−18.05	−3.25	−6.05	0.92	5.75	−13.77	−23.77

## Data Availability

The data presented in this study are available contained within the article.

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
