# Peer review of "Mitigating the Adverse Effects of Polychlorinated Biphenyl Derivatives on Estrogenic Activity via Molecular Modification Techniques"

_ijerph, 2021, doi:10.3390/ijerph18094999_

Round 1

Reviewer 1 Report

The work presented in this reviewable article concerns the study of the attenuation of the undesirable effects of polychlorinated biphenyl derivatives on estrogenic activity via molecular modification techniques (3D-QSAR model combined with molecular docking technology to study in detail the differential expression mechanism of the anti-estrogenic activity of PCB molecules before and after modification). Authors show that the adjustment of the human microenvironment can indirectly affect expression of estrogenic activity of PCBS molecules exposed to the human environment and mitigate the risk of estrogenic activity of residual PCBS molecules in the human body.

I had proposed to accept the manuscript for publication after minor revisions (reformatting tables (3 & 7) for clearer reading and correcting text editing errors).... The authors having responded to my comments, I therefore confirm the publication proposal after a thorough proofreading to correct typographical errors in particular.

Reviewer 2 Report

The manuscript has merit to be published. In order to decrease estrogenic activity, this study provides relevant information toward the design of new molecules based on modification of PCBs´ molecular structure using docking techniques, which could contribute to reduce harmful effects on biodiversity and human health. However, authors should consider to include experimental information based on similar works to validate the conclusions.

Reviewer 3 Report

Comments:

In this study, the authors applied QSAR to explore the mechanism of the change in estrogenic activities of PCBs derivatives. This is a well-written manuscript and worth publishing. However, it is a bit difficult to follow the flow of this article. I think readers may have the same issue. I may suggest adding a screening procedure as a figure in the revised manuscript. Other than this, I have several minor comments as follows.

  1. Line 11: please remove “further”.
  2. Lines 23-24: should these two sentences be combined?
  3. Line 40: in both humans? Something might be wrong here. Please check.
  4. Lines 49-50: it is better to remove “the only”.
  5. Line 61: it says several countries have banned the production of PCBs. Is this true? As I know, PCBs have been listed in the Stockholm Convention on Persistent Organic Pollutants, calling for a worldwide elimination and ban.
  6. Figure 1: can the authors please add a legend to describe the yellow, red or blue pattern in the figure? Also, the authors should mention the chemical is PCB 44 in the caption.
  7. Figure 2. The resolution of this figure is so low that I can not recognize the words.
  8. Table 7: are these PCB-44-1 to PCB-44-7 are the seven PCBs derivatives described in the previous paragraphs. If so, please use their full names.
  9. Please check the references. The order of references is not matched with those in the main text.

Author Response

This manuscript is a resubmission of an earlier submission. The following is a list of the peer review reports and author responses from that submission.

Round 1

Reviewer 1 Report

In this paper, the authors use a molecular binding model to show that modifications to the structure of PCBs can alter their ability to bind to proteins.

However, it is difficult to understand the purpose of this research because the background information is not sufficiently described. Are the authors considering incorporating these techniques in their processing technology? Or were modifieds structure known to be produced by metabolism?

Furthermore, I did not understand the toxicological implications of experimentally modifying the structure of PCBs to change their ability to bind to proteins.

What were the criteria used to select CB44 and its modifications for this study? Do they have any significance, such as generation in vivo?

“2:1 ratio, and used the Minimize molecular program under SYBYL-X 2.0 to optimize the molecular mechanics of each molecule”

Does this mean 3-fold cross validation?

Table 2 Please spell out the abbreviation for each indicator

2.3: “The target molecule PCB-44 and its 7 types of”, 3.2 “In this study, we used the Surflex-Dock module in SYBYL-X2.0 to bind the 7 types of”

Why are only these seven substances included in this analysis?

Reviewer 2 Report

The work presented in this reviewable article concerns the study of the attenuation of the undesirable effects of polychlorinated biphenyl derivatives on estrogenic activity via molecular modification techniques (3D-QSAR model combined with molecular docking technology to study in detail the differential expression mechanism of the anti-estrogenic activity of PCB molecules before and after modification). Authors show that the adjustment of the human microenvironment can indirectly affect expression of estrogenic activity of PCBS molecules exposed to the human environment and mitigate the risk of estrogenic activity of residual PCBS molecules in the human body.

The works presented are original and represent consistent results, fully and fully exploited in the discussions opened by the authors. However, for a better understanding and reading by a larger scientific audience, it would be interesting for the structures or at least the generic structures to be presented in a classic way in a Table / Figure summarizing these structures.

I therefore propose to accept the manuscript for publication after minor revisions (reformatting of tables (3 & 7) for clearer reading and correction of text editing errors). It would be good if a table / lexicon includes all the acronyms.

Round 2

Reviewer 1 Report

Point 1, 2: Thank you for your answer.

But, I think these points need to be mentioned in the text.

Point 3: I had missed the description of the previous study, thank you.

But I cannot find the sentence about criterion. Doesn't it need to be included in the objective to change more than 30%?

Point 4: I think this point needs to be described in detail in the manuscript.

Point 6: I couldn't find the description of the criteria authors answered in the manuscript.

Also, is the criterion of a change of 30% or more supported by previous studies as a meaningful value?

I think these points need to be mentioned in the text.